# Improving Sampling Accuracy of Stochastic Gradient MCMC Methods via Non-uniform Subsampling of Gradients

## Abstract

Common Stochastic Gradient MCMC methods approximate gradients by stochastic ones via uniformly subsampled data points. A non-uniform subsampling scheme, however, can reduce the variance introduced by the stochastic approximation and make the sampling of a target distribution more accurate. For this purpose, an exponentially weighted stochastic gradient approach (EWSG) is developed to match the transition kernel of a non-uniform-SG-MCMC method with that of a batch-gradient-MCMC method. If needed to be put in the importance sampling (IS) category, EWSG can be viewed as a way to extend the IS+SG approach successful for optimization to the sampling setup. EWSG works for a range of MCMC methods, and a demonstration on Stochastic-Gradient 2nd-order Langevin is provided. In our practical implementation of EWSG, the non-uniform subsampling is performed efficiently via a Metropolis-Hasting chain on the data index, which is coupled to the sampling algorithm. The fact that our method has reduced local variance with high probability is theoretically analyzed. A non-asymptotic global error analysis is also presented. As a practical implementation contains hyperparameters, numerical experiments based on both synthetic and real world data sets are provided, to both demonstrate the empirical performances and recommend hyperparameter choices. Notably, while statistical accuracy has improved, the speed of convergence, with appropriately chosen hyper-parameters, was empirically observed to be at least comparable to the uniform version, which renders EWSG a practically useful alternative to common variance reduction treatments.

## 1 Introduction

Many MCMC methods use physics-inspired evolution such as Langevin dynamics (Brooks et al., 2011) to utilize gradient information for exploring posterior distributions over continuous parameter space efficiently. However, gradient-based MCMC methods are often limited by the computational cost of computing the gradient on large data sets. Motivated by the great success of stochastic gradient methods for optimization, stochastic gradient MCMC methods (SG-MCMC) for sampling have also been gaining increasing attention. When the accurate but expensive-to-evaluate batch gradients in a MCMC method are replaced by computationally cheaper estimates based on a subset of the data, the method is turned to a stochastic gradient version. Classical examples include SG (overdamped) Langevin Dynamics (Welling & Teh, 2011) and SG Hamiltonian Monte Carlo (Chen et al., 2014), all of which were designed for scalability suitable for machine learning tasks.

However, directly replacing the batch gradient by a (uniform) stochastic one without additional mitigation will generally cause a MCMC method to sample from a statistical distribution different from the target, because the transition kernel of the MCMC method gets corrupted by the noise of subsampled gradient. In general, the additional noise is tolerable if the learning rate/step size is tiny or decreasing. However, when large steps are used for better efficiency, the extra noise is non-negligible and undermines the performance of downstream applications such as Bayesian inference.

In this paper, we present a state-dependent non-uniform SG-MCMC algorithm termed Exponentially Weighted Stochastic Gradients method (EWSG), which continues the efforts of uniform SG-

MCMC methods for better scalability. Our approach is based on designing the transition kernel of a SG-MCMC method to match the transition kernel of a full-gradient-based MCMC method. This matching leads to non-uniform (in fact, exponential) weights that aim at capturing the entire state-variable distribution of the full-gradient-based MCMC method, rather than just providing unbiased gradient estimator or reducing its variance. When focusing on the variance, the advantage of EWSG is the following: recall the stochasticity of a SG-MCMC method can be decomposed into the intrinsic randomness of MCMC and the randomness introduced by gradient subsampling; in conventional uniform subsampling treatments, the latter randomness is independent of the former, and thus when they are coupled together, variances add up; EWSG, on the other hand, dynamically chooses the weight of each datum according to the current state of the MCMC, and thus the variances do not add up due to dependence. However, the gained accuracy is beyond reduced variance, as EWSG, when converged, samples from a distribution close to the invariant distribution of the full-gradient MCMC method (which has no variance of the 2nd type), because its transition kernel (of the corresponding Markov process) is close to that of the full-gradient-MCMC method. This is how better sampling accuracy can be achieved.

Our main demonstration of EWSG is based on 2nd-order Langevin equations (a.k.a. inertial, kinetic, or underdamped Langevin), although it works for other MCMC methods too (e.g., Sec.F,G). To concentrate on the role of non-uniform SG weights, we will work with constant step sizes only. The fact that EWSG has locally reduced variance than its uniform counterpart is rigorously shown in Theorem 3, and a global non-asymptotic analysis of EWSG is given in Theorem 4 to quantify its convergence properties and demonstrate the advantage over its uniform SG counterpart.

A number of experiments on synthetic and real world data sets, across downstream tasks including Bayesian logistic regression and Bayesian neural networks, are conducted to validate our theoretical results and demonstrate the effectiveness of EWSG. In addition to improved accuracy, the convergence speed was empirically observed, in a fair comparison setup based on the same data pass, to be comparable to its uniform counterpart when hyper-parameters are appropriately chosen. The convergence (per data pass) was also seen to be clearly faster than a classical Variance Reduction (VR) approach (note: for sampling, not optimization), and EWSG hence provides a useful alternative to VR. Additional theoretical investigation of EWSG convergence speed is provided in Sec. I.

Terminology-wise, $\nabla V$ will be called the full/batch-gradient, $n\nabla V_I$ with random $I$ will be called stochastic gradient (SG), and when $I$ is uniform distributed it will be called a uniform SG/subsampling, otherwise non-uniform. When uniform SG is used to approximate the batch-gradient in underdamped Langevin, the method will be referred to as (vanilla) stochastic gradient underdamped Langevin dynamics (SGULD/SGHMC)[1], and it serves as a baseline in experiments.

## 2 RELATED WORK

**Stochastic Gradient MCMC Methods**    Since the seminal work of SGLD (Welling & Teh, 2011), much progress (Ahn et al., 2012; Patterson & Teh, 2013) has been made in the field of SG-MCMC. Teh et al. (2016) theoretically justified the convergence of SGLD and offered practical guidance on tuning step size. Li et al. (2016) introduced a preconditioner and improved stability of SGLD. We also refer to Maclaurin & Adams (2015) and Fu & Zhang (2017) which will be discussed in Sec.5. While these work were mostly based on 1st-order (overdamped) Langevin, other dynamics were considered too. For instance, Chen et al. (2014) proposed SGHMC, which is closely related to 2nd-order Langevin dynamics (Bou-Rabee & Sanz-Serna, 2018; Bou-Rabee et al., 2018), and Ma et al. (2015) put it in a more general framework. 2nd-order Langevin was recently shown to be faster than the 1st-order version in appropriate setups (Cheng et al., 2018b;a) and began to gain more attention.

**Variance Reduction**    For **optimization**, vanilla SG methods usually find approximate solutions quickly but the convergence slows down when an accurate solution is needed (Bach, 2013; Johnson & Zhang, 2013). SAG (Schmidt et al., 2017) improved the convergence speed of stochastic gradient methods to linear, which is the same as gradient descent methods with full gradient, at the expense of large memory overhead. SVRG (Johnson & Zhang, 2013) successfully reduced this memory overhead. SAGA (Defazio et al., 2014) furthers improved convergence speed over SAG and SVRG. For

---

[1]SGULD is the same as the well-known SGHMC with $\hat{B} = 0$, see (Chen et al., 2014, Eq (13) and section 3.3) for details. To be consistent with existing literature, we will refer SGULD as SGHMC in the sequel.

**sampling**, Dubey et al. (2016) applied VR techniques to SGLD (see also (Baker et al., 2019; Chatterji et al., 2018)). However, many VR methods have large memory overhead and/or periodically use the whole data set for gradient estimation calibration, and hence can be resource-demanding.

EWSG is derived based on matching transition kernels of MCMC and improves the accuracy of the entire distribution rather than just the variance. However, it does have a consequence of variance reduction and thus can be implicitly regarded as a VR method. When compared to the classic work on VR for SG-MCMC (Dubey et al., 2016), EWSG converges faster when the same amount of data pass is used, although its sampling accuracy is below that of VR for Gaussian targets (but well above vanilla SG; Sec.5.1). In this sense, EWSG and VR suit different application domains: EWSG can replace vanilla SG for tasks in which the priority is speed and then accuracy, as it keeps the speed but improves the accuracy; on the other hand, VR remains to be the heavy weapon for accuracy-demanding scenarios. Importantly, EWSG, as a generic way to improve SG-MCMC methods, can be combined with VR too (e.g., Sec.G); thus, they are not exclusive or competitors.

**Importance Sampling (IS)** IS employs nonuniform weights to improve SG methods for **optimization**. Traditional IS uses fixes weights that do not change along iterations, and the weight computation requires prior information of gradient terms, e.g., Lipschitz constants of gradient (Needell et al., 2014; Schmidt et al., 2015; Csiba & Richtárik, 2018), which are usually unknown or difficult to estimate. Adaptive IS was also proposed in which the importance was re-evaluated at each iteration, whose computation usually required the entire data set per iteration and may also require information like the upper bound of gradient (Zhao & Zhang, 2015; Zhu, 2016).

For **sampling**, it is not easy to combine IS with SG (Fu & Zhang, 2017); the same paper is, to our knowledge, the closest to this goal and will be compared with in Sec.5.3. EWSG can be viewed as a way to combine (adaptive) IS with SG for efficient sampling. It require no oracle about the gradient, nor any evaluation over the full data set. Instead, an inner-loop Metropolis chain maintains a random index that approximates a state-dependent non-uniform distribution (i.e. the weights/importance).

## 3 UNDERDAMPED LANGEVIN: THE BACKGROUND OF A MCMC METHOD

Underdamped Langevin Dynamics (ULD) is
$$\begin{cases} d\boldsymbol{\theta} & = \boldsymbol{r}dt \\ d\boldsymbol{r} & = -(\nabla V(\boldsymbol{\theta}) + \gamma\boldsymbol{r})dt + \sigma d\boldsymbol{W} \end{cases} \tag{1}$$

where $\boldsymbol{\theta}, \boldsymbol{r} \in \mathbb{R}^d$ are state and momentum variables, $V$ is a potential energy function which in our context (originated from cost minimization or Bayesian inference over many data) is the sum of many terms $V(\boldsymbol{\theta}) = \sum_{i=1}^n V_i(\boldsymbol{\theta})$, $\gamma$ is a friction coefficient, $\sigma$ is intrinsic noise amplitude, and $\boldsymbol{W}$ is a standard $d$-dimensional Wiener process. Under mild assumptions on the potential $V$ (Pavliotis, 2014), Langevin dynamics admits a unique invariant distribution $\pi(\boldsymbol{\theta}, \boldsymbol{r}) \sim \exp\left(-\frac{1}{T}(V(\boldsymbol{\theta}) + \frac{\|\boldsymbol{r}\|^2}{2})\right)$ and is in many cases geometric ergodic. $T$ is the temperature of system determined via the fluctuation dissipation theorem $\sigma^2 = 2\gamma T$ (Kubo, 1966).

The main reason for considering ULD rather than overdamped one is that ULD can converge faster than overdamped Langevin, in particular in high-dimension space (e.g.,Cheng et al. (2018b;a); Tao & Ohsawa (2020)). Like the overdamped version, numerical integrators for ULD with well captured statistical properties of the continuous process have been extensively investigated (e.g, Roberts et al. (1996); Bou-Rabee & Owhadi (2010)), and both the overdamped and underdamped integrators are friendly to derivations that will allow us to obtain explicit expressions of the non-uniform weights.

## 4 MAIN WORK

### 4.1 AN ILLUSTRATION OF NON-OPTIMALITY OF UNIFORM SUBSAMPLING

In many applications, cases where data size $n$ is larger than dimension $d$ are not uncommon. In such cases, $\{\nabla V_i\}_{i=1,2,\cdots,n} \subset \mathbb{R}^d$ are linearly dependent and hence it is likely that there exist probability distributions $\{p_i\}_{i=1,2,\cdots,n}$ other than the uniform one such that the gradient estimate is unbiased. This opens up the door to develop non-uniform subsampling schemes (weights may be $\boldsymbol{\theta}$ dependent),

which can help reduce introduced additional variance while maintaining unbiasedness. In fact, in a reasonable setup, it turns out an optimal way of subsampling gradients, is far from being uniform:

**Theorem 1** *Suppose given $\boldsymbol{\theta} \in \mathbb{R}^d$, the errors of SG approximation $\boldsymbol{b}_i = n\nabla V_i(\boldsymbol{\theta}) - \nabla V(\boldsymbol{\theta}), 1 \leq i \leq n$ are i.i.d. absolutely continuous random vectors with possibly-$\boldsymbol{\theta}$-dependent density $p(x|\boldsymbol{\theta})$. Define $\boldsymbol{p} \in \mathbb{R}^n$ as a sparse vector if the number of non-zero entries in $\boldsymbol{p}$ is no greater than $d+1$. Then with probability 1, the optimal probability distribution $\boldsymbol{p}^\star$ that is unbiased and minimizes the trace of the covariance of $n\nabla V_I(\boldsymbol{\theta})$, i.e. $\boldsymbol{p}^\star$ which solves the following, is a sparse vector.*

$$\min_{\boldsymbol{p}} \mathrm{Tr}(\mathbb{E}_{I \sim \boldsymbol{p}}[\boldsymbol{b}_I \boldsymbol{b}_I^T]) \quad s.t. \ \mathbb{E}_{I \sim \boldsymbol{p}}[\boldsymbol{b}_I] = \boldsymbol{0}, \tag{2}$$

Despite the sparsity of $\boldsymbol{p}^\star$, which seemingly suggests one only needs at most $d+1$ gradient terms per iteration when using SG methods, it is not practical because $\boldsymbol{p}^\star$ requires solving the linear programming problem (2) in Theorem 1, for which an entire data pass is needed. Nevertheless, this result shows uniform SG can be far from optimal and motivates us to propose an exponentially weighted stochastic gradient method, which has reduced local variance with high probability and at the same time remains efficiently implementable without necessarily using all the data per parameter update.

## 4.2 EXPONENTIALLY WEIGHTED STOCHASTIC GRADIENT

MCMC methods or Markov processes in general are characterized by their transition kernels. In traditional SG-MCMC methods, uniform SG is used, which is completely independent of the intrinsic randomness of MCMC methods (e.g. diffusion in ULD), as a result, the transition kernel of SG-MCMC method can be quite different from that with full gradient. Therefore, it is natural to ask - is it possible to couple these two originally independent randomness so that the transition kernels can be better matched and the sampling accuracy can be hence improved?

Consider Euler-Maruyama (EM) discretization[2] of Equation (1):

$$\begin{cases} \boldsymbol{\theta}_{k+1} &= \boldsymbol{\theta}_k + \boldsymbol{r}_k h \\ \boldsymbol{r}_{k+1} &= \boldsymbol{r}_k - (\nabla V(\boldsymbol{\theta}_k) + \gamma \boldsymbol{r}_k)h + \sigma\sqrt{h}\boldsymbol{\xi}_{k+1} \end{cases} \tag{3}$$

where $h$ is step size and $\boldsymbol{\xi}_{k+1}$'s are i.i.d. $d$-dimensional standard Gaussian random variables. Denote the transition kernel of EM discretization with full gradient by $P^{EM}(\boldsymbol{\theta}_{k+1}, \boldsymbol{r}_{k+1}|\boldsymbol{\theta}_k, \boldsymbol{r}_k)$. If $\nabla V(\boldsymbol{\theta}_k)$ is replaced by a weighted SG $n\nabla V_{I_k}(\boldsymbol{\theta}_k)$, where $I_k$ is the index chosen to approximate full gradient and has p.m.f $\mathbb{P}(I_k = i) = p_i$, denote the transition kernel by $\tilde{P}^{EM}(\boldsymbol{\theta}_{k+1}, \boldsymbol{r}_{k+1}|\boldsymbol{\theta}_k, \boldsymbol{r}_k)$. It turns out that we can choose $p_i$ smartly to match the two transition kernels:

**Theorem 2** *Denote $\boldsymbol{x} = \frac{\boldsymbol{r}_{k+1} - \boldsymbol{r}_k + h\gamma\boldsymbol{r}_k}{\sigma\sqrt{h}}$ and $\boldsymbol{a}_i = \frac{\sqrt{h}\nabla V_i(\boldsymbol{\theta}_k)}{\sigma}$. If we set*

$$p_i = \hat{Z}^{-1} \exp\left\{ -\frac{\|\boldsymbol{x} + \sum_{j=1}^n \boldsymbol{a}_j\|^2}{2} + \frac{\|\boldsymbol{x} + n\boldsymbol{a}_i\|^2}{2} \right\} \tag{4}$$

*where $\hat{Z}$ is a normalization constant, then the two transition kernels are identical, i.e.,*

$$\tilde{P}^{EM}(\boldsymbol{\theta}_{k+1}, \boldsymbol{r}_{k+1}|\boldsymbol{\theta}_k, \boldsymbol{r}_k) = P^{EM}(\boldsymbol{\theta}_{k+1}, \boldsymbol{r}_{k+1}|\boldsymbol{\theta}_k, \boldsymbol{r}_k)$$

We refer to this choice of $p_i$ Exponentially Weighted Stochastic Gradient (**EWSG**). Note the idea of designing non-uniform weights of SG-MCMC to match the transition kernel of full gradient can be suitably applied to a wide class of gradient-based MCMC methods; for example, Sec.F shows how EWSG can be applied to Langevin Monte Carlo (overdamped Langevin), and Sec.G shows how it can be combined with VR. Therefore, EWSG complements a wide range of SG-MCMC methods.

Thm.2 establishes the advantage of EWSG over vanilla SG, as this ideal version reproduces the distribution of a full-gradient MCMC method. As a special but commonly interested accuracy measure, the smaller variance of EWSG is now shown with high probability[3]:

---

[2]EM is not the most accurate or robust discretization, see e.g., (Roberts et al., 1996; Bou-Rabee & Owhadi, 2010), but since it may still be the most used method, demonstrations in this article will be based on EM. The same idea of EWSG can easily apply to most other discretizations such as GLA (Bou-Rabee & Owhadi, 2010).

[3]'With high probability' but not almost surely because Thm.3 is not tight as it can handle more general weights, including not only the ideal EWSG weights (4) but also their appropriate approximations.

**Theorem 3** *Assume* $\{\nabla V_i(\boldsymbol{\theta})\}_{i=1,2,\cdots,n}$ *are i.i.d random vectors and* $|\nabla V_i(\boldsymbol{\theta})| \leq R$ *for some constant* $R$ *almost surely. Denote the uniform distribution over* $[n]$ *by* $\boldsymbol{p}^U$, *the exponentially weighted distribution by* $\boldsymbol{p}^E$, *and let* $\Delta = \mathrm{Tr}[cov_{I \sim \boldsymbol{p}^E}[n\nabla V_I(\boldsymbol{\theta})|\boldsymbol{\theta}] - cov_{I \sim \boldsymbol{p}^U}[n\nabla V_I(\boldsymbol{\theta})|\boldsymbol{\theta}]]$. *If* $\boldsymbol{x} = \mathcal{O}(\sqrt{h})$, *we have* $\mathbb{E}[\Delta] < 0$, *and* $\exists C > 0$ *independent of* $n$ *or* $h$ *such that for any* $\epsilon > 0$, $\mathbb{P}(|\Delta - \mathbb{E}[\Delta]| \geq \epsilon) \leq 2 \exp\left(-\frac{\epsilon^2}{nCh^2}\right)$.

It is not surprising that less non-intrinsic local variance correlates with better global statistical accuracy, which will be made explicit and rigorous in Section 4.4.

### 4.3 PRACTICAL IMPLEMENTATION

In EWSG, the probability of each gradient term is $p_i = \hat{Z}^{-1} \exp\left\{-\frac{\|\boldsymbol{x}+\sum_{j=1}^n \boldsymbol{a}_j\|^2}{2} + \frac{\|\boldsymbol{x}+n\boldsymbol{a}_i\|^2}{2}\right\}$. Although the term $\|\boldsymbol{x} + \sum_{j=1}^n \boldsymbol{a}_j\|^2/2$ depends on the full data set, it is shared by all $p_i$'s and can be absorbed into the normalization constant $\hat{Z}^{-1}$ (we still included it explicitly due to the needs of analyses in proofs); unique to each $p_i$ is only the term $\|\boldsymbol{x} + n\boldsymbol{a}_i\|^2/2$. This motivates us to run a Metropolis-Hasting chain over the possible indices $i \in \{1, 2 \cdots, n\}$: at each inner-loop step, a proposal of index $j$ is uniformly drawn, and then accepted with probability

$$P(i \to j) = \min\left\{1, \exp\left(\frac{\|\boldsymbol{x}+n\boldsymbol{a}_j\|^2}{2} - \frac{\|\boldsymbol{x}+n\boldsymbol{a}_i\|^2}{2}\right)\right\};\tag{5}$$

if accepted, the current index $i$ will be replaced by $j$. When the chain converges, the index will follow the distribution given by $p_i$. The advantage is, we avoid passing through the entire data sets to compute each $p_i$, but yet the index will still sample from the non-uniform distribution efficiently.

In practice, we often only perform $M = 1$ step of the Metropolis index chain per integration step, especially if $h$ is not too large. The rationale is, when $h$ is small, the outer iteration evolves slower than the index chain, and as $\theta$ does not change much in, say, $N$ outer steps, effectively $N \times M$ inner steps take place on almost the same index chain, which makes the index r.v. equilibrate better. Regarding the larger $h$ case (where the efficacy of local variance reduction via non-uniform subsampling is more pronounced; see e.g., Thm.4), $M = 1$ may no longer be the optimal choice, but improved sampling with large $h$ and $M = 1$ is still clearly observed in various experiments (Sec.5).

Another hyper-parameter is $\boldsymbol{x}$, because $p_i$ essentially depends on the future state $\boldsymbol{\theta}_{k+1}$ via $\boldsymbol{x}$, which we do not know, and yet we'd like to avoid expensive nonlinear solves. Therefore, in our experiments, we choose $\boldsymbol{x} = \frac{\sqrt{h}\gamma \boldsymbol{r}_k}{\sigma}$. That corresponds to a deterministic maximum likelihood estimation of $\boldsymbol{r}_{k+1} = \boldsymbol{r}_k$, which is a sufficient (but not necessary) condition for mimicking the statistical equilibrium at which $\boldsymbol{r}_{k+1}$ and $\boldsymbol{r}_k$ are equal in distribution. This approximation turned out to be a good one in all our experiments with medium $h$ and $M = 1$. Because it is only an approximation, when $h$ is large, the method still introduces extra variance (smaller than that caused by vanilla stochastic gradient variant, though), and larger $M$ may actually decrease the accuracy of sampling.

Sec.5.1 further investigates hyperparameters and if they affect our non-asymptotic theory in Sec.4.4.

EWSG algorithm is summarized in Algorithm 1. For simplicity of notation, we restrict the description to mini batch size $b = 1$, but an extension to $b > 1$ is straightforward. See Sec. E in appendix. EWSG has reduced variance but does not completely eliminate the nonintrinsic noise created by stochastic gradient due to these approximations. A small bias was also created by these approximations, but its effect is dominated by the variance effect (see Sec.4.4). In practice, if needed, one can combine EWSG with other variance reduction technique to further improve accuracy. We showcase how EWSG can be combined with SVRG in Sec.G of appendix.

### 4.4 NON-ASYMPTOTIC ERROR BOUND

The generator $\mathcal{L}$ of diffusion process (1) is $\mathcal{L} = (\boldsymbol{r}^T \nabla_{\boldsymbol{\theta}} - (\gamma \boldsymbol{r} + \nabla V(\boldsymbol{\theta}))^T \nabla_{\boldsymbol{r}} + \gamma \Delta_{\boldsymbol{r}})$. Let $\boldsymbol{X} = (\boldsymbol{\theta}^T, \boldsymbol{r}^T)^T \in \mathbb{R}^{2d}$. Given a test function $\phi(\boldsymbol{x})$, its posterior average is $\bar{\phi} = \int \phi(\boldsymbol{x})\pi(\boldsymbol{x})d\boldsymbol{x}$, and we approximate it by time average of samples $\widehat{\phi}_K = \frac{1}{K}\sum_{k=1}^K \phi(\boldsymbol{X}_k^E)$, where $\boldsymbol{X}_k^E$ is the sample path given by EM integrator. A useful tool in weak convergence analysis for SG-MCMC is the Poisson

---

**Algorithm 1** EWSG

---

**Input:** {the number of data terms $n$, gradient functions $V_i(\cdot), i = 1, 2, \cdots, n$, step size $h$, the number of data passes $K$, index chain length $M$, friction and noise coefficients $\gamma$ and $\sigma$}
Initialize $\boldsymbol{\theta}_0, \boldsymbol{r}_0$ (arbitrarily, or use an informed guess)
**for** $k = 0, 1, \cdots, \lceil \frac{Kn}{M+1} \rceil$ **do**
   $i \leftarrow$ uniformly sampled from $1, \cdots, n$, compute and store $n\nabla V_i(\boldsymbol{\theta}_k)$
   $I \leftarrow i$
   **for** $m = 1, 2, \cdots, M$ **do**
      $j \leftarrow$ uniformly sampled from $1, \cdots, n$, compute and store $n\nabla V_j(\boldsymbol{\theta}_k)$
      $I \leftarrow j$ with probability in Equation 5
   **end for**
   Evaluate $\tilde{V}(\boldsymbol{\theta}_k) = nV_I(\boldsymbol{\theta}_k)$
   Update $(\boldsymbol{\theta}_{k+1}, \boldsymbol{r}_{k+1}) \leftarrow (\boldsymbol{\theta}_k, \boldsymbol{r}_k)$ via one step of Euler-Maruyama integration using $\tilde{V}(\boldsymbol{\theta}_k)$
**end for**

---

equation $\mathcal{L}\psi = \phi - \bar{\phi}$ (Mattingly et al., 2010; Vollmer et al., 2016; Chen et al., 2015). The solution $\psi$ characterizes the difference between test function $\phi$ and its posterior average $\bar{\phi}$.

We now bound the error (in mean square distance between arbitrary test observables) for SG underdamped Langevin algorithms (the bound applies to both EWSG and other methods e.g., SGHMC):

**Theorem 4** *Assume $\mathbb{E}[\|\nabla V_i(\boldsymbol{\theta}_k^E)\|^l] < M_1, \mathbb{E}[\|\boldsymbol{r}_k^E\|^l] < M_2, \forall l = 1, 2, \cdots, 12, \forall i = 1, 2, \cdots, n$ and $\forall k \geq 0$. Assume the Poisson equation solution $\psi$ and up to its 3rd-order derivatives are uniformly bounded $\|D^l\psi\|_\infty < M_3, l = 0, 1, 2, 3$. Then $\exists$ constant $C = C(M_1, M_2) > 0$, s.t.*

$$\mathbb{E}(\widehat{\phi}_K - \bar{\phi})^2 \leq C\left(\frac{1}{T} + \frac{h}{T}\frac{\sum_{k=0}^{K-1}\mathbb{E}[\text{Tr}[\text{cov}(n\nabla V_{I_k}|\mathcal{F}_k)]]}{K} + h^2\right) \tag{6}$$

*where $T = Kh$ is the corresponding time in the underlying continuous dynamics, $I_k$ is the index of the datum used to estimate gradient at $k$-th iteration, and $cov(n\nabla V_{I_k}|\mathcal{F}_k)$ is the covariance of stochastic gradient at $k$-th iteration conditioned on the current sigma algebra $\mathcal{F}_k$ in the filtration.*

**Remark:** Mattingly et al. (2010) only discusses the batch gradient case, whereas our theory has additional (non-uniform) stochastic gradient. Vollmer et al. (2016); Chen et al. (2015) studied the effect of stochastic gradient, but the SG considered there did not use state-dependent weights, which would destroy several martingales used in their proofs. In addition, our result incorporates the effects of both local bias and local variance of a SG approximation. Unlike in Mattingly et al. (2010) but like in Vollmer et al. (2016); Chen et al. (2015), our state space is not the compact torus but $\mathbb{R}^d$. The time average $\widehat{\phi}_K$, to which our results apply, is a commonly used estimator, particularly when simulating a single Markov chain. Techniques in Cheng et al. (2018b); Dalalyan & Karagulyan (2017) might be useful to further bound difference between the law of $\boldsymbol{X}_k$ and the target distribution $\pi$.

Variance and bias of the SG approximation were reflected in the 2nd and 3rd term in the above bound, although the 3rd term also contains a contribution from the numerical integration error. Note the 2nd term is generally larger than the 3rd due to its lower order in $h$, which means reducing local variance can improve sampling accuracy even if at the cost of introducing a small bias. Since EWSG has a smaller local variance than uniform SG (Thm.3, as a special case of improved overall statistical accuracy), its global performance is also favorable.

## 5 EXPERIMENTS

In this section, the proposed EWSG algorithm will be compared with SGHMC, SGLD (Welling & Teh, 2011), as well as several recent popular SG-MCMC methods, including FlyMC (Maclaurin & Adams, 2015), pSGLD (Li et al., 2016), CP-SGHMC (Fu & Zhang, 2017) (closest method to IS for sampling by SG-MCMC) and SVRG-LD (Dubey et al., 2016) (overdamped Langevin improved by VR). Sec. 5.1 is a detailed empirical study of EWSG on simple models, with comparison and implication of two important hyper-parameters $M$ and $\boldsymbol{x}$, and verification of the non-asymptotic theory (Thm.4). Sec. 5.2 demonstrates the effectiveness of EWSG for Bayesian logistic regression on a large-scale data set. Sec. 5.3 shows the performance of EWSG on Bayesin Neural Network

(BNN) model. BNN only serves as a high-dimensional, multi-modal test case and we do not intend to compare Bayesian against non-Bayesian neural nets. As FlyMC requires a tight lower bound of likelihood, known for only a few cases, it will only be compared against in Sec. 5.2 where such a bound is obtainable. CP-SGHMC requires heavy tuning on the number of clusters which differs across data sets/algorithms, so it will only be included in the BNN example, for which the authors empirically found a good hyper parameter for MNIST (Fu & Zhang, 2017). SVRG-LD is only compared in Sec. 5.1, because SG-MCMC methods converge in one data pass in Sec. 5.2, rendering control-variate based VR technique inapplicable, and it was suggested that VR leads to poor results for deep models (e.g., Sec.5.3) (Defazio & Bottou, 2019)

For fair comparison, all algorithms use constant step sizes and are allowed fixed computation budget, i.e., for $L$ data passes, all algorithms are only allowed to call gradient function $nL$ times. All experiments are conducted on a machine with a 2.20GHz Intel(R) Xeon(R) E5-2630 v4 CPU and an Nvidia GeForce GTX 1080 GPU. If not otherwise mentioned, $\sigma = \sqrt{2\gamma}$ so only $\gamma$ needs specification, the length of the index chain is set $M = 1$ for EWSG and the default values of two hyper-parameters required in pSGLD are set $\lambda = 10^{-5}$ and $\alpha = 0.99$, as suggested in Li et al. (2016).

## 5.1 GAUSSIAN EXAMPLES

Consider sampling from a simple 2D Gaussian whose potential function is $V(\boldsymbol{\theta}) = \sum_{i=1}^{n} V_i(\boldsymbol{\theta}) = \sum_{i=1}^{n} \frac{1}{2}\|\boldsymbol{\theta} - \boldsymbol{c}_i\|^2$. We set $n = 20$ and randomly sample $\boldsymbol{c}_i$ from a two-dimensional standard normal $\mathcal{N}(\boldsymbol{0}, I_2)$. Due to the simplicity of $V(\boldsymbol{\theta})$, we can write the target density analytically and will use KL divergence to measure the difference between the target distribution and generated samples.

For each algorithm, we generate 10000 independent realizations for empirical estimation. All algorithms are run for 30 data passes with minibatch size of 1. Step size is tuned from $5 \times \{10^{-1}, 10^{-2}, 10^{-3}, 10^{-4}\}$ and $5 \times 10^{-3}$ is chosen for SGLD and pSGLD, $5 \times 10^{-2}$ for SGHMC and EWSG and $5 \times 10^{-4}$ for SVRG-LD. SGHMC and EWSG use $\gamma = 10$. Results are shown in Fig. 1a and EWSG outperforms SGHMC, SGLD and pSGLD in terms of accuracy. Note SVRG-LD has the best accuracy[4] but the slowest convergence, and that is why EWSG is a useful alternative to VR: its light-weight suits situations with limited computational resources better.

When simulating a gradient-based Markov chain, large step size generally reduces autocorrelation time[5], yet leads to large discretization error. Figure 1b shows at the same autocorrelation time, EWSG achieves smaller error than SGHMC, which demonstrates the effectiveness of EWSG.

Figure 1c shows the performance of several possible choices of the hyper-parameter $\boldsymbol{x}$, including the proposed option $\boldsymbol{x} = \sqrt{h}\gamma\boldsymbol{r}_k/\sigma$, $\boldsymbol{x} = \boldsymbol{0}$, $\boldsymbol{x} = \boldsymbol{1}$ and $\boldsymbol{x} = (-1 + h\gamma)\boldsymbol{r}_k/\sigma\sqrt{h}$ (corresponds to setting $\boldsymbol{r}_{k+1} = \boldsymbol{0}$). The result shows that the proposed option performs significantly better than other alternatives and we suggest to use it by default.

Another important hyper-parameter in EWSG is $M$. As the length of index chain $M$ increases, the distribution approaches the distribution given by Equation (4), which by Theorem 4 introduces some bias but also reduces variance. The tradeoff is clearly manifested in Figure 1d and 1e. $M$, regarded as a hyper-parameter, controls this tradeoff and generally requires tuning to determine the best value.

The optimal value of $M$ depends on the number of minibatches $\frac{n}{b}$. Large $\frac{n}{b}$ implies each minibatch is relatively small, and suggests large variance in stochastic gradient estimation of each minibatch and hence a strong need to reduce variance, see the illustration in Figure 1i. For example, if we increase the number of data in Gaussian example to $n = 50$ or $100$ and keep mini batch size $b = 1$, $M$ that gives the best final accuracy increases to 19 in Figure 1g and 1h. In our experiments, we empirically observe the choice $M = 1$ usually gives reasonably good performance and we fix it as the default choice. See Sec. H.3 for additional experiments on tuning $M$ and the batch size.

In Figure 1a, we see EWSG converges slightly slower than uniform SG. This is because in each iteration, EWSG consumes $M + 1$ times the data used by uniform SG, and hence in a fair comparison with fixed computation budget, EWSG runs $M$ times iterations fewer than uniform SG. However, EWSG can still achieve comparable speed of convergence with appropriately chosen hyper parame-

---

[4]For Gaussians, mean and variance completely determine the distribution, so appropriately reduced variance leads to great accuracy for the entire distribution.

[5]Autocorrelation time is defined as $\tau = 1 + \sum_{s=0}^{\infty} \rho_s$, where $\rho_s$ is the autocorrelation at time lag $s$.

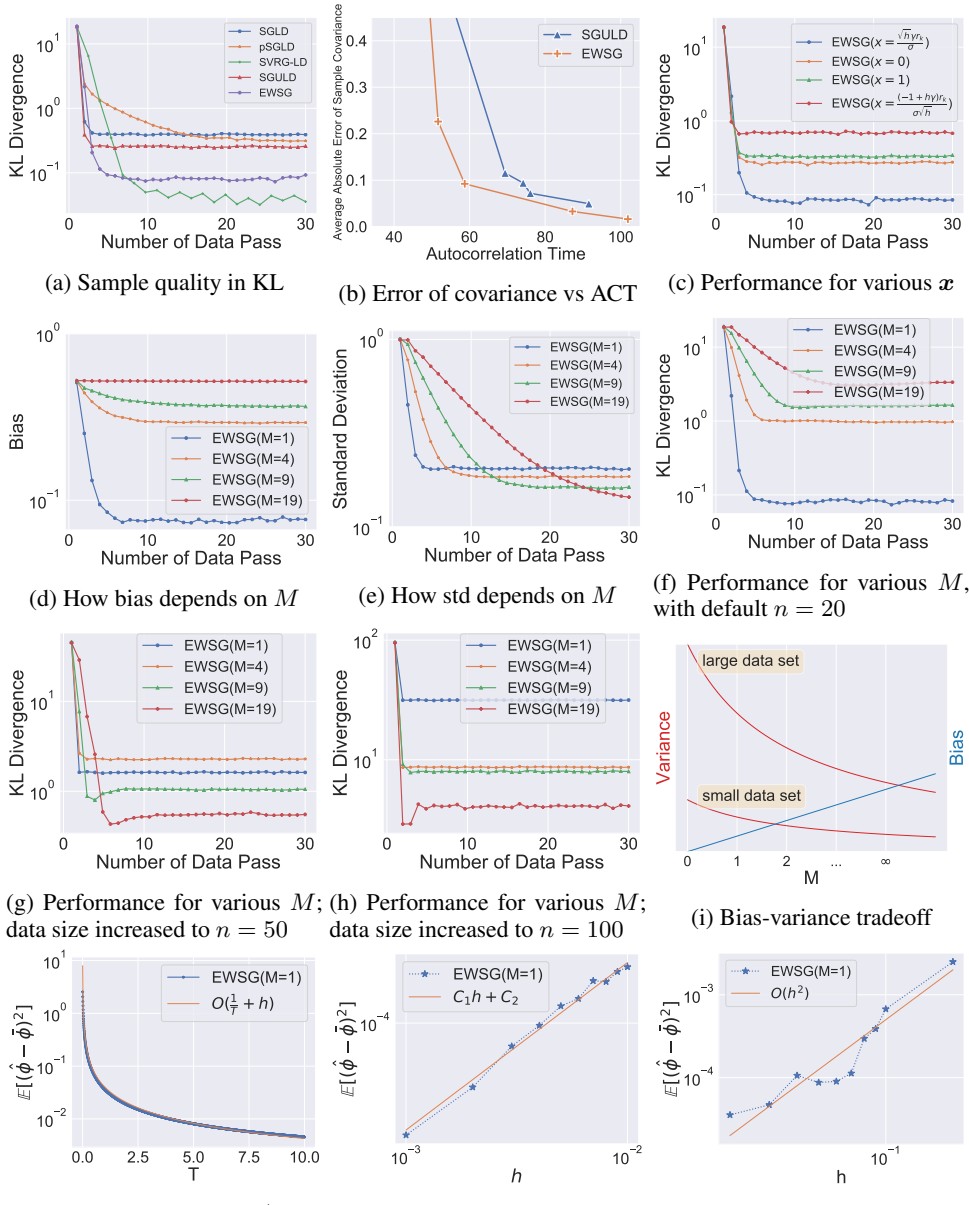

(a) Sample quality in KL

(b) Error of covariance vs ACT

(c) Performance for various $\boldsymbol{x}$

(d) How bias depends on $M$

(e) How std depends on $M$

(f) Performance for various $M$, with default $n = 20$

(g) Performance for various $M$; data size increased to $n = 50$

(h) Performance for various $M$; data size increased to $n = 100$

(i) Bias-variance tradeoff

(j) MSE against time $T$ ($1^{\text{st}}$ term in Eq. (6))

(k) MSE against step size $h$ with fixed finite $T$ ($2^{\text{nd}}$ term in Eq. (6))

(l) MSE against step size $h$ with $T \approx \infty$ ($3^{\text{rd}}$ term in Eq. (6))

Figure 1: Sampling from Gaussian target

ters; e.g., if uniform SG uses minibatch size $b$, then one can use a smaller minibatch size (e.g. $\frac{b}{M+1}$) for EWSG to ensure both algorithms consume the same number of gradient calls per iteration and hence run the same number of iterations. See Sec. H.3 for more empirical studies on this.

As approximations are used in Alg.1, it is natural to ask if Thm.4 still applies. We empirically investigate this question (using $M = 1$ and variance as the test function $\phi$). Eq.(6) in Thm.4 has a nonasymptotic error bound consisting of three parts, namely an $\mathcal{O}(\frac{1}{T})$ term corresponding to the convergence at the continuous limit, an $\mathcal{O}(h/T)$ term coming from the SG variance, and an $\mathcal{O}(h^2)$ term due to bias and numerical error. Fig.1j plots the mean squared error (MSE) against time $T = Kh$ to confirm the 1st (and the 2nd) term. Fig.1k plots the MSE against $h$ in the small $h$ regime (so that the 2nd term dominates the 3rd) to confirm that the 2nd term scales like $\mathcal{O}(h)$ with a fixed $T$. For the 3rd term in Eq. (6), we run sufficiently many iterations to ensure all chains are well-mixed, and Fig.11 confirms the final MSE to scale like $\mathcal{O}(h^2)$ even for large $h$ (as the 2nd term vanishes due to $T \to \infty$). In this sense, despite the approximations introduced by the practical

| Method | SGLD | pSGLD | SGHMC | EWSG | FlyMC |
|---|---|---|---|---|---|
| Accuracy(%) | $75.282 \pm 0.079$ | $75.079 \pm 0.094$ | $75.272 \pm 0.069$ | $\mathbf{75.293 \pm 0.045}$ | $75.165 \pm 0.079$ |
| Log Likelihood | $-0.525 \pm 0.000$ | $-0.527 \pm 0.000$ | $-0.525 \pm 0.000$ | $\mathbf{-0.523 \pm 0.000}$ | $\mathbf{-0.523 \pm 0.001}$ |

Table 1: Accuracy and log likelihood of BLR on test data after one data pass (mean $\pm$ std).

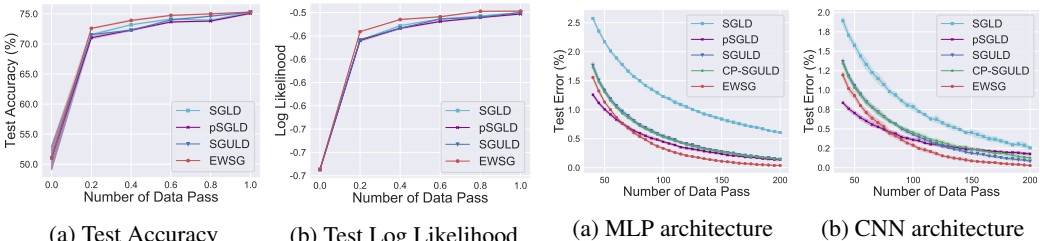

(a) Test Accuracy    (b) Test Log Likelihood    (a) MLP architecture    (b) CNN architecture

Figure 2: BLR learning curve      Figure 3: BNN learning curve. Shade: 1 std.

implementation, the performance of Algorithm 1 is still approximated by Theorem 4, even when $M = 1$, and Theorem 4 provides reasonable guidelines for practical use of the EWSG algorithm.

## 5.2 BAYESIAN LOGISTIC REGRESSION (BLR)

Consider binary classification based on problablistic model $p(y_k = 1|\boldsymbol{x}_k, \boldsymbol{\theta}) = 1/(1 + \exp(-\theta^T \boldsymbol{x}_k))$. We set Gaussian prior $N(\boldsymbol{0}, 10I_d)$ for $\boldsymbol{\theta}$ and experiment with the Covertype data set [6] (581,012 data points, 54 features). We use 80% of data for training and the rest for testing.

ULD based algorithms use $\gamma = 50$. After tuning, we set step sizes as $\{1, 3, 0.02, 5, 5\} \times 10^{-3}$ for SGHMC, EWSG, SGLD, pSGLD and FlyMC. All algorithms are run for one data pass, with minibatch size of 50. 200 independent samples are drawn from each algorithm to estimate statistics.

Results are in Fig. 2a and 2b and Table 1. EWSG outperforms others, except for log likelihood being comparable to FlyMC, which is an *exact* MCMC method.

## 5.3 BAYESIAN NEURAL NETWORK (BNN)

Bayesian inference is compelling for deep learning (Wilson, 2020). Two popular architecture of neural nets are experimented – multilayer perceptron (MLP) and convolutional neural nets (CNN). In MLP, a hidden layer with 100 neurons followed by a softmax layer is used. In CNN, we use standard network configuration with 2 convolutional layers followed by 2 fully connected layers (Jarrett et al., 2009). Both convolutional layers use $5 \times 5$ convolution kernel with 32 and 64 channels, $2 \times 2$ max pooling layers follow immediately after convolutional layer. The last two fully-connected layers each has 200 neurons. We set the standard normal as prior for all weights and bias.

We test algorithms on the MNIST data set, consisting of 60000 training data and 10000 test data, each datum is a $28 \times 28$ gray-scale image with one of the ten possible labels (digits $0 \sim 9$). For ULD based algorithms , we set friction coefficient $\gamma = 0.1$ in MLP and $\gamma = 1.0$ in CNN. In MLP, the step sizes are set $h = \{4, 2, 2\} \times 10^{-3}$ for EWSG, SGHMC and CP-SGHMC, and $h = \{0.001, 1\} \times 10^{-4}$ for SGLD and pSGLD, via grid search. For CP-SGHMC , we use K-means with 10 clusters to preprocess the data set. In CNN, the step sizes are set $h = \{4, 2, 2\} \times 10^{-3}$ for EWSG, SGHMC and CP-SGHMC, and $h = \{0.02, 8\} \times 10^{-6}$ for SGLD and pSGLD, via grid search. All algorithms use minibatch size of 100 and are run for 200 data passes. For each algorithm, we generate 100 independent samples to estimate posterior distributions and make prediction accordingly.

The learning curve of training accuracy is shown in Figure 3a and 3b. We find EWSG consistently improve over its uniform counterpart (i.e. SGHMC) and CP-SGHMC (an approximate IS SG-MCMC). Moreover, EWSG also outperforms two standard benchmarks SGLD and pSGLD. The improvement over baseline on MNIST data set is comparable to some of the early works (Chen et al., 2014; Li et al., 2016). More results on this experiment can be found in Sec. H.2.

---

[6]https://archive.ics.uci.edu/ml/datasets/covertype

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
