# OpenReview forum: "Improving Sampling Accuracy of Stochastic Gradient MCMC Methods via Non-uniform Subsampling of Gradients"
_ICLR.cc/2021/Conference — Reject_

### Official Review · AnonReviewer2 · 2020-10-18
**The paper proposes an innovative sampling scheme to construct the minibatches used in stochastic gradient algorithms.**

**Rating:** 6
**Confidence:** 3

**Review:**

##########################################################################
Summary:

The paper proposes an alternative to the uniform sampling scheme used for constructing mini-batches in stochastic gradient sampling algorithms. The proposed scheme, called Exponentially Weighted Stochastic Gradient (EWSG), is devised such its transition kernel matches that of the batch gradient descent. The proposed scheme is shown to achieve better results than the uniform sampling one.

##########################################################################
Reason for score:

I'm oscillating between accepting the paper and rejecting it. Overall there are good merits to the paper, however there are some aspects on which I would like to have some further clarifications from the authors.

##########################################################################
Pros:

1. The paper is overall well written with a proper formulation and justification of the problem it addresses. The proposed solution is analysed both theoretically and empirically and the results show the advantage of the proposed method with respect to the uniform sampling scheme.

2. The subject the paper addresses is of importance to the ML community and I believe the advances brought by the paper are sufficient to warrant its acceptance.

##########################################################################
Cons:

1. I have some concerns regarding the practical implementation of the proposed scheme in general. Overall, I appreciated the discussion about the choice of the hyper-parameters. However, my concerns stems from the fact that you have two intertwining Markov chains, one for the quantity we wish to perform inference on and one for the indices. From figure 1(a) we can see that there is a slower convergence of the EWSG method with respect to the SGLD one, though this slower converge is compensated by the better accuracy. My question is whether in all practical situations I will eventually reach a better accuracy than SGLD within an acceptable computational budget?
On the same note, in the paper you deal only with the case of one item per mini-batch. You address in the appendix the extension to mini-batch sizes greater than one. However, when we consider mini-batch sizes different than one, for some mini-batch sizes the number of possible combinations is greater than the number of elements in the data set. The sampling over the indexes will lead to sampling the "best" mini-batches, however would the advantage of the proposed method still be retained within a fixed computational budget in such a scenario given that there are more possibilities that need exploring?

2. Another issue stems from the fact that the method requires access to the full dataset, or at least to be able to retrieve items from it. There are situations in which the full dataset is distributed across different computational nodes, thus on one node I would have only a limited subset of data. Assuming we can recombine the results from each node to get a global estimate, would it still be beneficial to run your proposed method on each node such that to improve accuracy or in such a case running SGLD would suffice? This question is related to the assumption that you make at the beginning of section 4.1 that n >> d and I'm just wondering if there are any advantages of using your proposed method when the number of data items is still greater than the dimension, but not significantly.

3. I have a minor issue with the plots in figure 1: the values on the ordinate axis are sometimes very difficult to see and read, you could increase the font given that there still is space available before reaching the margins of the page.

---

> ### Author Response · Authors · 2020-11-25
> **Response to AnonReviewer2**
>
> Thanks very much for the valuable and insightful comments. Glad to hear that our subject is of importance to the community and that the paper is overall well written. Please see the itemized reply below:
>
> 1.
>
>  EWSG does not necessarily converge slower than the uniform SG version. The reviewer was absolutely right in pointing out the slow-down in Figure 1(a), and we very much appreciate the comment as it helped us realize that our presentation was confusing and gave us an opportunity to clarify. In short, whether there is a slow-down depends on the hyperparmeters, and our experiments with Bayesian neural networks are examples where EWSG improves accuracy without converging slower.
>
> Here are more details. The slow-down in Fig.1(a) is due to the fact that both EWSG and uniform SG use the same minibatch size $b$. However, in each iteration, EWSG consumes $M+1$ minibatches because of the index chain whereas uniform SG only requires 1 minibatch, and hence in a fair comparison with the same computation budget, EWSG runs $M$ times iterations fewer than the uniform SG. In general, if uniform SG uses minibatch size $b$, one can use minibatch size $\frac{b}{M+1}$ for EWSG so that both algorithms require the same number of gradient calls per iteration and run the same number of iterations. In this case, the convergence speed of EWSG will be comparable to that of uniform SG. We empirically compared EWSG (with minibatch size $\frac{b}{M+1}$) and uniform SG (with minibatch size $b$) in the Bayesian neural network setup (see Table 3 in Section H in supplementary materials). The experiment results show that even with a smaller minibatch size,  EWSG still consistently outperforms its uniform counterpart.
>
> 2.
>
> Distributed sampling is a very interesting and important suggestion, despite that it could be a little beyond the scope of this paper. We read a good number of articles on dynamics-based MCMC methods published in leading conferences/journals, but have not seen much relevant discussion, and we thus think this great idea may create a new venue of investigations and very much appreciate the suggestion. At this moment, we feel that the first question to ask may not be how distributed sampling could be different for MCMC based on uniform subsampling (traditional methods) and EWSG subsampling (this paper), but rather, whether there is any subsampling scheme that allows a good partition and distribution of the data. Meanwhile, we also think that if there is already a good distribution schedule for uniform subsampling, the analogous schedule for EWSG may not necessarily be different, at least when $h$ is small, because EWSG weights are actually $\exp(\mathcal{O}(h))/Z=(1+\mathcal{O}(h))/Z$. We hope we could be allowed to consider distributed sampling in the future as it deserves an independent, careful study.
>
>  We appreciate the comment on the ``"assumption" $n \gg d$, as it points out another location where our clarity could be improved. It is NOT required for EWSG, but just used as a special example to demonstrate the suboptimality of uniform subsampling in a concrete way (Theorem 1). In our Bayesian neural network experiment, there are $n=50000$ training data, but $d > 78400$ parameters, and in this $n < d$ case, EWSG still outperforms all (uniform subsampling) benchmarks.
>
> 3.
>
> Thanks for the helpful comment. The axis ticks as well as sub-figures in Figure 1 were indeed too small and they are now enlarged.

---

### Official Review · AnonReviewer4 · 2020-10-28
**Recommendation to Reject**

**Rating:** 4
**Confidence:** 3

**Review:**

This paper proposes a non-uniform sampling method for stochastic gradient minibatches for SG-MCMC. By sampling the indices of the stochastic gradients according to a parameter-specific (exponentially weighted) non-uniform distribution, the paper shows that it exactly matches the transition kernel of full batch gradient MCMC (for underdamped langevin dynamics) in Theorem 2. Although this exponentially weight distribution is intractable, the paper presents an approximate sampler in Algorithm 1 (that both uses a 1-step Metropolis Hastings approximation and a deterministic approximation for the $x$ term $r_{k+1} = r_k$). Finally the paper compares Algorithm 1 with other SGMCMC methods in a small synthetic Gaussian example, bayesian logistic regression on Covertype data, and a Bayesian neural network on MNIST.

The idea to non-uniformly sample the random stochastic gradient index to match SGULD is clever. The paper would benefit from additional editing: the notation and presentation of ideas is unclear + difficult to parse in some sections.

This paper should be rejected. Although the particular non-uniform random sampling scheme is novel, it is not clear how the two approximations (MH and $x$) in Algorithm 1 affect the error of the sampler in practice. Although the experiments show some benefit of Algorithm 1 over vanilla SGMCMC, they are not strongly convincing or exhaustive and do not analyze the error introduced in Algorithm 1. I believe with significant editing and additional experiments, this paper would be more competitive, but as it stands, it should be rejected.

Although Theorems 1-4 may be interesting for the non-uniform sampling scheme for stochastic gradient indices (Eq 4), in practice, Algorithm 1 is different due to the two approximations. In particular, I am concerned about the approximation for $x$ (especially since the definition of $x$ seems circular). Although it may appear that the approximation $r_{k+1} = r_k$ works well for your experiments, this may not be the case in general. Clarity on when this approximation is justified and when it is not would help build confidence in this approximation. I would have liked to have seen how the performance of Algorithm 1 compared to the theory (Theorem 3 or 4) for the theoretical sampler using (Eq 4) or the full batch "ground truth" MCMC (Eq 3).

---

> ### Author Response · Authors · 2020-11-25
> **Response to AnonReviewer4**
>
> We deeply appreciate the reviewer’s valuable comments. Following the great suggestion, we added a comparison of the performances of Algorithm 1 and the theory (Thm.4), which allowed us to much better confirm the validity of the approximations introduced in Sec.4.3. This made the paper much stronger, and hopefully could turn the doubts away.
>
> More precisely, a series of comparisons were made to compare the practical implementation (with the approximations) with the theoretical bound, term by term. They correspond to newly added Fig.1(j,k,l) and a new paragraph toward the end of pg.8. The result was that the recommended approximations ($x=0$,$M=1$) did not damage the theoretical bound, in the sense that
>
> - when $T$ is not too large such that $1/T$ dominates $h^2$, the mean squared error (MSE) decays like $\mathcal{O}(\frac{1}{T})$ with fixed $h$ (1st and 2nd terms in the theoretical bound);
> - with fixed $T$, the MSE changes with $h$ in a linear way, for $h$ that dominates $h^2$ (2nd term in the theoretical bound);
> - when the chain is well-mixed (i.e., $T\approx\infty$), the final MSE is $\mathcal{O}(h^2)$ (3rd term in the theoretical bound).
>
> In terms of incorporating the effects of hyperparameters into our non-asymptotic global error bound, regretfully we feel the mathematical tools are lacking. Although definitely imperfect, may we suggest that Thm.4 is already rather nontrivial and in some sense the SOTA? Previous best bound known to us was based on unbiased gradient estimators and thus cannot reflect the additional contribution from bias. Extending the heavy machinery of Poisson techniques for Markov semigroups, we managed to obtain a bound that accounts for both bias and variance, which allowed us to quantify their trade-off. Our result is rather general as common assumptions such as (strong-)log-concavity of target distribution are not needed. Quantifying how variance and bias simultaneously contribute to the global error is, as far as we know, new. The main stream is to reduce the variance while maintaining the mean, but we treat the entire distribution as a whole and thus both the mean and variance can be off (but only by a little).
>
> Note the previous SOTA bounds did not (tightly) reflect the roles of hyperparameters either, but were nevertheless regarded as important contributions. Sometimes the contribution of one of those great articles was solely theoretical, without a new algorithm OR comprehensive numerical verifications. Therefore, we sincerely hope the reviewers could consider that our paper has a new idea at both the continuous and the algorithmic levels, and a substantial theoretical analysis, and a detailed empirical study.
>
> To see an aspect of the theoretical challenges, suppose we need a tight characterization of the effect of $M$. To support the $M=1$ recommendation, we then need non-asymptotics of non-asymptotics, as we have two Markov processes nested together, and the continuous family of the inner Markov chain non-asymptotics (as the inner chain is nonlinearly indexed by the outer MCMC state variable) need to be uniform and just for 1 inner step. This means, e.g., any warm start popular in the MCMC literature is disallowed. As far as we know, even the nonasymptotics of SGD in a nonconvex setting with tight characterizations of the hyperparameters is still an active research frontier. In this sense, some help to let the idea grow in a series of studies, instead of killing it in infancy, would be deeply appreciated.
>
> Additional itemized responses:
>
> 1.
>
> We recommended the hyperparameter value $x=0$ based on the following heuristic: when at equilibrium, both $r_k$ and $r_{k+1}$ are centered Gaussians. Since $x=(r_{k+1}-r_k+h\gamma r_k)/\sigma/\sqrt{h}$ in the ideal EWSG, ideal $x$ would be Gaussian with 0 mean, and thus its maximum likelihood is achieved at $x=0$.
>
> The reviewer is absolutely correct that this is not proved, and no matter how many empirical tests we conduct, there is no guarantee that this choice will work well for all experiments. This is certainly a drawback of our work. Meanwhile, we feel it is common for current machine learning research to have hyperparameters not completely characterized by theory but supported by heuristics and empirical evidence. For example, even the choice of learning rate in SGD is still being studied despite of years of research and many milestone results.
>
> 2.
>
> We were sorry to hear that our notation and presentation were unclear and difficult to parse in some sections. We re-edited the paper for improvement.

---

### Official Review · AnonReviewer3 · 2020-10-28
**The authors provide the novel connection between stochastic gradient and importance sampling.**

**Rating:** 5
**Confidence:** 4

**Review:**

##  Summary of the paper
The authors focused on the fact that there are two types of stochasticity in SG-MCMCs, one is the intrinsic randomness of MCMC and the other is the randomness introduced by gradient subsampling. Then the authors proposed coupling these randomnesses so that the final variance is reduced. Experimental results support that the proposed coupling seems promising.

## Strong and weak points of the paper
### Strong points
- Provided a new connection between importance sampling and stochastic gradient.
- Provided an asymptotic analysis for the variance reduction in Theorem 4, which seems novel variance analysis.

### Weak points
- I am not sure the approximations introduced in Sec 4.3 are valid theoretically and numerically. I commend these points in the below.

## Rating
- Clarity: Well written, easy to read, although I did not check the proof of Theorem 4.
- Correctness: I did not check all the proof in detail.
- Novelty: The idea seems very interesting and important in the community.

## Comments and Questions
- Q) the authors introduced several approximations, e.g., setting M=1,  setting x as the equibirum points, to implement the index sampling efficiently. How those approximations affect the final variance theoretically ? Especially, although the authors provided the variance analysis in Theorem 4, I think it is unclear how these approximations affect the upper bound of the variance in Eq.(6) explcitely.

- Q) could you explain why the approximation of $r_{k+1}=r_k$ in Sec 4.3 is valid and when this approximation is not valid ? At first sight I thought that this equilibrium condition is definitely wrong at an early stage of the underdamped Langevin dynamics.

- Q) In Fig 1.d to 1.f, the authors studied the impact of M. It seems that the larger M are, the worse results are obtained. Is this correct ? I thought that increasing M means increasing the number of Markov chains for the index sampling and it should improve the performance.

---

> ### Author Response · Authors · 2020-11-25
> **Response to AnonReviewer3**
>
> We deeply appreciate the reviewer's valuable comments which greatly help us improve our paper; for example, we now have a much better demonstration of the validity of the approximations introduced in Sec 4.3. It is also great to hear an affirmation of our novelty, clarity, and the importance of the subject. Here are itemized responses:
>
> 1.
>
> Indeed, Thm.4 only provides a bound for the ideal EWSG algorithm, and the approximations (due to hyperparameters, namely a finite $M$ and a heuristic $x$) used in the practical implementation are not reflected in this bound. In the revision, we added substantial experiments to verify that the bound in Thm.4 still holds numerically, term by term, even after the introduction of these approximations. Kindly see the newly added Fig.1(j,k,l) and the text on pg.8.
>
>
> In terms of theoretically incorporating the effects of hyperparameters into our bound, regretfully our attempts have not been successful. Thm.4 is rather nontrivial and in some sense already the SOTA, as the previous best bound known to us was based on unbiased gradient estimators and thus cannot reflect the additional contribution from bias. Extending the heavy machinery of Poisson equation technique, we managed to obtain a bound that accounts for both bias and variance, which allowed us to quantify their trade-off. Our result is rather general as common assumptions such as (strong-)log-concavity of target distribution are not needed. Quantifying how variance and bias simultaneously contribute to the global error is, as far as we know, new.
>
> Note the previous SOTA bounds did not (tightly) reflect the roles of hyperparameters either, but were nevertheless regarded as important contributions. Sometimes the contribution of one of those great articles was solely theoretical, without a new algorithm OR comprehensive numerical verifications. Therefore, we sincerely hope the reviewers could consider that our paper has a new idea at both the continuous and the algorithmic levels, and a substantial theoretical analysis, and a detailed empirical study.
>
> To see an aspect of the theoretical challenges, suppose we need a tight characterization of the effect of $M$, we then need non-asymptotics of non-asymptotics, as we have two Markov processes nested together, and the continuous family of the inner Markov chain non-asymptotics (as the inner chain is nonlinearly indexed by the outer MCMC state variable) need to be uniform and just for 1 inner step. This means, e.g., any warm start popular in the MCMC literature is disallowed. As far as we know, even the nonasymptotics of SGD in a nonconvex setting with tight characterizations of the hyperparameters is still an active research frontier. In this sense, some help to let the idea grow in a series of studies, instead of killing it in infancy, would be deeply appreciated.
>
> 2.
>
> Our recommended x was based on the following heuristic: when at equilibrium, both $r_k$ and $r_{k+1}$ are centered Gaussians.  In the ideal EWSG, $x$ would be Gaussian with 0 mean, and thus its MLE is achieved at 0. Hence the intuitive choice. The reviewer is absolutely correct that in the early phase of underdamped Langevin, this approximation may fail. Still, in almost all cases we experimented, $x=0$ led to good performances, and a comparison of several heuristic values of x is provided in Fig.1c. We thus reported $x=0$.
>
> 3.
>
> The reviewer's observations about the impact of $M$ were absolutely correct and greatly appreciated, as it made us realize that our original presentation was misleading. We improved the presentation in Fig.1 and the text. In short, the confusion was due to: small data effect, and that larger $M$ gives fewer outer iterations per data pass.
>
> Here is more explanation. In the $M\to\infty$ limit, there will be a small bias due to the heuristic $x$, and Fig.1(d) reflects this trend. Fig.1(e) shows, however, that one of the gains is reduced variance. This has a cost too, as larger $M$ gave slower convergence in a fair comparison. Therefore, there is a variance/bias trade-off which is quantified by Thm.4 and illustrated by newly added Fig.1(i).
>
> Does this mean that larger $M$ is actually bad? Fig.1(f) in the original version implied so and it was our fault having created this inaccurate implication. For large $M$, EWSG will have increased bias and reduced variance. Whether this trade-off is worthwhile depends on how much variance is introduced by SG: consider the ratio between the data set size n and the minibatch size b. Larger $\frac{n}{b}$ implies larger variance of SG and a stronger need to reduce variance. Our original Fig.1(f) used $b=1, n=20$ and this gives a variance not large enough, and hence the bias-variance trade-off of EWSG is not favorable when M is large. We repeat the experiment with $b = 1$ but larger data size $n = 50, 100$. The results clearly show (newly added Fig.1(g,h)) that larger M gives better accuracy as the larger variance is more worth trading.

---

### Decision · Program_Chairs · 2021-01-07
**Final Decision**

**Decision:**

Reject

**Comment:**

This paper describes a non-uniformly weighted version of SGMCMC, combining aspects of SG methods and importance sampling. The idea is interesting and novel, but unfortunately the authors have not made a compelling case for the resulting algorithm being a practical addition to the literature. The experimental analysis is not particularly compelling, and there are key concerns raised about practical implementation, and about the validity of the approximations raised. I hope that the authors will continue along this interesting line of work and add additional explorations of the approximations and improved experimental analysis.